# Exploring how people with dementia can be best supported to manage long-term conditions: a qualitative study of stakeholder perspectives

Jessica Laura Rees ,[1] Alexandra Burton,[1] Kate R Walters ,[2] Monica Leverton,[1] Penny Rapaport,[1] Ruminda Herat Gunaratne,[1] Julie Beresford-Dent,[3] Claudia Cooper[1]

¹Division of Psychiatry, University College London, London, UK
²Primary Care and Population Health, University College London, London, UK
³Centre for Applied Dementia Studies, University of Bradford, Bradford, UK

**Correspondence to**
Jessica Laura Rees;
jessica.rees@ucl.ac.uk

## ABSTRACT

**Objectives** To explore how the self-management of comorbid long-term conditions is experienced and negotiated by people with dementia and their carers.
**Design** Secondary thematic analysis of 82 semi-structured interviews.
**Setting** Community settings across the United Kingdom.
**Participants** 11 people with dementia, 22 family carers, 19 health professionals and 30 homecare staff.
**Results** We identified three overarching themes: (1) *The process of substituting self-management*: stakeholders balanced the wishes of people with dementia to retain autonomy with the risks of lower adherence to medical treatments. The task of helping a person with dementia to take medication was perceived as intermediate between a personal care and a medical activity; rules about which professionals could perform this activity sometimes caused conflict. (2) *Communication in the care network*: family carers often communicated with services and made decisions about how to implement medical advice. In situations where family carers or homecare workers were not substituting self-management, it could be challenging for general practitioners to identify changes in self-management and decide when to intervene. (3) *Impact of physical health on and from dementia*: healthcare professionals acknowledged the inter-relatedness of physical health and cognition to adapt care accordingly. Some treatments prescribed for long-term conditions were perceived as unhelpful when not adapted to the context of dementia. Healthcare professionals and homecare workers sometimes felt that family carers were unable to accept that available treatments may not be helpful to people with dementia and that this sometimes led to the continuation of treatments of questionable benefit.
**Conclusion** The process of substituting self-management evolves with advancement of dementia symptoms and relies on communication in the care network, while considering the impact on and from dementia to achieve holistic physical health management. Care decisions must consider people with dementia as a whole, and be based on realistic outcomes and best interests.

## INTRODUCTION

Globally, the prevalence of dementia is predicted to rise to 131.5 million by 2050

## Strengths and limitations of this study

► This was a large study involving 82 in-depth interviews with participants.
► Participants were from diverse backgrounds and a range of locations across the UK.
► Interviews included the perspective of homecare staff in addition to people with dementia, family carers and healthcare professionals.
► A limitation to secondary data analysis is that the data were originally collected to explore how to support independence at home for people living with dementia.
► Family carers discussed multi-morbidity most frequently so are relatively over-represented in the analysis.

compared with 46.8 million in 2015.[1] Over 470 000 people on General Practice Registers in England have a formal diagnosis of dementia.[2] Almost 8 in every 10 people with dementia live with another chronic long-term disease or condition.[3] The health needs within this population are complex.[4] The most prevalent multi-morbidities: hypertension, diabetes and coronary heart disease,[5] can increase the risk of dementia, while conditions such as stroke, epilepsy and depression are also associated with increased dementia risk.[3] The optimal management of physical health may help to prevent hospitalisation and slow cognitive decline.[6 7]

Self-management is an integral and lifelong task for people with long-term conditions.[8] The symptoms of dementia, including loss of memory, disorientation and executive dysfunction, frequently reduce an individual's ability to organise and plan self-care.[9] The most commonly described impact is the diminishing ability to administer medication safely.[10 11] Most people with dementia need

help to manage care for comorbid long-term conditions—usually from a family carer or homecare worker, or both.[12–14] Partnership between this care network at home and primary care is critical for the management of long-term conditions in dementia.[15]

Research has highlighted a need for guidance on how to tailor and individualise care for people with dementia and long-term conditions.[16 17] Clinical guidelines aim to promote evidence-based best practice but tend to focus on single conditions, so may not reflect difficulties in managing multi-morbidity in dementia.[18–20] Understanding the lived experiences of stakeholders is critical to the development of relevant clinical guidelines, including considering interactions between comorbidities on the ability of an individual to self-manage their care.[21] The first step towards service development includes understanding the care needs of people living with dementia and coexisting long-term conditions.[4] A recent review of the literature found limited evidence about how self-management and support by other stakeholders, such as homecare workers, can support long-term condition management in dementia.[15]

This paper aims to explore the experiences of people living with dementia, family carers, healthcare professionals and homecare staff, to identify how the management of long-term conditions is best supported in dementia.

## METHODS

The New Interventions for Independence in Dementia (NIDUS) Study (ISRCTN11425138/ISRCTN99460116) conducted qualitative, semi-structured interviews with people with dementia, family carers, health and social care professionals and homecare staff. The primary analysis of this qualitative data explored how people with dementia can be supported to live as independently as possible in their own homes,[22] including a separate analysis on the experiences of South Asian family carers.[23] We carried out a secondary analysis of the transcripts to explore the management of long-term conditions for people living at home with dementia.

### Recruitment

People with dementia and family carers were recruited through three UK National Health Service (NHS) memory services, private homecare services, an Alzheimer's Society Experts by Experience group and Twitter. Health and social care professionals were recruited through NHS memory services, social services and clinical academics with links to University College London. For the homecare staff interviews, 10 homecare agencies were recruited across England and managers, office support staff and homecare workers were invited to participate. Purposive sampling ensured our sample of people with dementia and family carers was diverse in terms of age, gender and ethnicity; for family carers, their relationship to care recipients; and for professionals, their professional roles and experiences with supporting people with dementia.

### Procedure

Participants were invited to take part in semi-structured qualitative interviews between April and September 2018. All participants gave written informed consent. Interviews lasted on average 1 hour and were conducted in participants' homes, workplaces or at university offices. Interviews followed a semi-structured topic guide which explored how people with dementia live independently at home and what support they need to do so. They were audio-recorded and transcribed verbatim. Across stakeholders, slightly amended versions of the topic guide were used. The interview schedule for people with dementia and family carers included specific questions about how long-term physical or mental health conditions affected the person with dementia's ability to remain living independently at home.

### Data analysis

Secondary analysis involves using pre-existing qualitative data collected from previous research studies to investigate new or additional research questions.[24] We took an inductive thematic approach through focusing on meaning within the data to develop codes[25] using NVivo 12 software.[26]

First, the lead author highlighted information in transcripts relating specifically to long-term conditions in addition to dementia. A coding framework was developed with authors (CC, KRW, AB) based on line-by-line coding of highlighted sections (see online supplemental material). The authors systematically labelled codes based on meaningful fragments of transcripts. We met to agree on re-occurring codes, and the lead author applied the coding framework to all transcripts. After completed analysis of all interviews, we reviewed and refined the coding framework by discussing themes corresponding to the research question. We revisited codes looking at commonalities and differences in managing long-term conditions across accounts of all stakeholders for the thematic analysis.

### Patient and public involvement

An advisory group of family carers and health professionals provided feedback on the findings from the NIDUS qualitative study which was incorporated into the analysis. We did not directly involve patients and the public in this secondary analysis, but we have invited members of the advisory group to help us develop our dissemination strategy.

## RESULTS

Qualitative interviews were conducted with 11 people with dementia (mean age=78.6, SD=7.8), 22 family carers (mean age=57.7, SD=14.3), 19 health and social care professionals (mean age=41.4, SD=10.9) and 30 homecare

| Table 1 Characteristics of participant demographics (person with dementia/family carer) | | |
|---|---|---|
| | **Person with dementia** | **Family carer** |
| Characteristics | n | n |
| Gender | | |
| Female | 5 | 12 |
| Male | 6 | 10 |
| Ethnicity | | |
| White British | 8 | 9 |
| White other | 1 | 0 |
| Indian | 0 | 6 |
| Bangladeshi | 0 | 4 |
| Other | 2 | 3 |
| Living arrangements | | |
| Lives alone | 5 | 7 |
| Lives with relatives | 6 | 15 |
| Type of dementia | | |
| Alzheimer's disease | 3 | 8 |
| Vascular | 2 | 4 |
| Other | 2 | 4 |
| Not specified | 4 | 6 |
| Relation to person with dementia | | |
| Son/daughter | – | 11 |
| Spouse | – | 6 |
| Niece | – | 2 |
| Friend | – | 1 |
| Sibling | – | 1 |
| Daughter-in-law | – | 1 |

| Table 2 Characteristics of participant demographics (professional carers) | | |
|---|---|---|
| | **Homecare staff** | **Health and social care professional** |
| Characteristics | n | n |
| Gender | | |
| Female | 25 | 13 |
| Male | 5 | 6 |
| Ethnicity | | |
| White British | 22 | 9 |
| White other | 2 | 5 |
| Black/Black British | 5 | 0 |
| Asian/Asian British | 1 | 0 |
| Indian | 0 | 2 |
| Other | 0 | 3 |
| Professional role | | |
| Managerial | 7 | 0 |
| Homecare worker | 19 | 0 |
| Other care staff | 4 | 0 |
| Commissioner | 0 | 3 |
| Social worker | 0 | 2 |
| Dementia lead | 0 | 2 |
| Service manager | 0 | 2 |
| Psychologist | 0 | 2 |
| Support worker | 0 | 2 |
| General practitioner | 0 | 2 |
| Geriatrician | 0 | 1 |
| Nurse | 0 | 1 |
| Physiotherapist | 0 | 1 |
| Psychiatrist | 0 | 1 |

staff (mean age=48.3, SD=11.6). To respond to the needs and preferences of participants, eight interviews were dyadic involving people with dementia and family carers together. Five family carers were born outside the UK. Details of participant characteristics are presented in tables 1 and 2.

The lead author (JR) extracted information from interview transcripts about long-term conditions discussed across all stakeholders. Interviews with people with dementia (n=9) and family carer (n=22) explicitly spoke about long-term conditions and support needs. Healthcare professionals (n=14) and homecare staff (n=17) included information on specific long-term conditions, but often spoke more generally about medication management and holistic aspects of health for example, falls and mobility. Figure 1 presents a full overview of reported long-term conditions by people with dementia, family carers, health and social care professionals and homecare staff.

We identified three main themes that responded to our research objectives to explore the experiences of people with dementia, family carers, healthcare professionals and homecare staff in managing long-term conditions in dementia. These were: (1) the process of substituting self-management; (2) communication in the care network; and (3) impact on and from dementia (when managing physical health).

### The process of substituting self-management
#### Autonomy versus risk
People with dementia valued autonomy when managing their physical health. In the earlier stages of dementia, prompting enabled self-management and a sense of 'still carrying on'. The prospect of requiring more help was a concern to this person living with dementia:

> You see I'm praying and I'm hoping that I won't need carers because I want to rely on myself. But I know at some stage I won't be able to. I don't know when. Maybe my health will get worse, physically worse, and

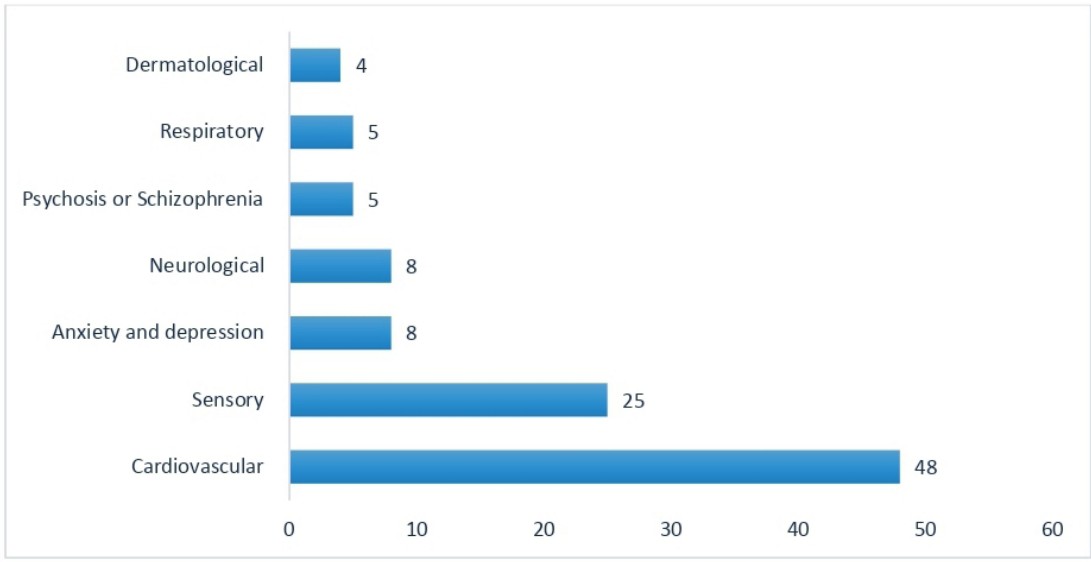

| Category | Long-term condition |
|---|---|
| **Mental health** | Anxiety and depression |
|  | Psychosis and Schizophrenia |
| **Sensory** | Visual impairment |
|  | Hearing impairment |
| **Musculoskeletal** | Arthritis |
|  | Osteoporosis |
| **Respiratory** | Asthma |
|  | Pulmonary fibrosis |
|  | Chronic obstructive pulmonary disease |
| **Cardiovascular** | Diabetes |
|  | Hypertension |
|  | Heart problems |
|  | Stroke |
| **Dermatological** | Eczema |
|  | Psoriasis |
|  | Dermatitis |

**Figure 1** Frequency of long-term conditions discussed in interviews.

this dementia will get worse and I will have to. (Person with Dementia_1, Arthritis and Hypertension)

Homecare workers acknowledged the wishes of people with dementia to retain decision-making control where possible and to be involved in their own care:

I say every time, this is for your memory, this is your thyroid, this is for your Addison's, this is for your blood pressure… And they say, oh thank God you told me because what you're doing is you're controlling them if you don't involve them. (Homecare Worker_1)

Family carers balanced the wishes of people living with dementia to retain autonomy as far as possible with the potential consequences of lower adherence to medical treatment. Family carers described their discomfort in providing care against the wishes of the person with dementia but acknowledged the importance of such tasks when memory impeded medication administration:

He then lost his eyesight in the left eye, due to the diabetes medication being missed. He was reporting

to my mother daily that he is consuming his medication. Of course, because of the dementia, his memory was deceiving him and telling him that he had done it, but this was a past memory. (Family Carer_1, Son)

### Prescriptions and role restrictions

Monitoring or administering medication was the most frequently discussed form of support for long-term condition management. While self-administration of medication is a routine part of long-term condition management, when a person with dementia required support of another person, this role occupied an intermediate position between a personal care activity and a medical task that required a nursing qualification. Rules around who administered medication could be complex and confusing:

Although we were told that the [home]carers are not allowed to medicate, because they're not insured. So, that's a difficult one. (Family Carer_2, Son)

One healthcare professional described the difficulties that could arise because of this when prescriptions changed:

It falls through when they're on, say steroids, like emergency steroids if they've got COPD or if they're on antibiotics for infection. Where the person with dementia will get really poorly because they're in boxes and [home]carers can't give out boxes; it has to be in a nomad [pre-packed medication]. (Community Mental Health Nurse)

### Communication in the care network

In healthcare settings, it was often the family carer who communicated with services and made decisions about how to implement medical advice. Being an implementer of medical advice but not the doctor, and the recipient of advice but not the patient, sometimes felt like a dilemma for family carers working as in partnership with people with dementia:

… I suppose at the end of the day I should go down and make an appointment to see the GP… to tell them that I feel she needs that done… but I don't want to go down and sort of feel that I'm… telling them what to do, coming heavy-handed on it either. (Family Carer_3, Sister)

Family carers experienced additional challenges in communication when developed relationships with primary care came to an end, as changes in practitioner's impacted continuity of care.

Where people with dementia experienced difficulties with communication, family carers offered support in healthcare appointments to ensure correct reporting of symptoms.

…she would tell them things that were factually incorrect and they would believe that they were correct. Because they just thought she might be a bit old and her literacy wasn't great… But she was on drugs that were negatively affecting her …she was on something that were actually negative to her heart. (Family Carer_4, Son)

Regardless of the presence of a proxy, participants with dementia highlighted the importance of continued acknowledgement within appointments:

It was, I think that's the problem, some people, because you've got diagnosed with it they think, oh he can't digest this, we'll talk to the relative. (Person with Dementia_2, Visual Impairment)

In situations where family carers or homecare workers were not substituting self-management, it could be challenging for general practitioners to identify changes in self-management and decide when to intervene:

But I'm not quite sure that the mechanisms are very good for picking up on that, you know, the bit where they're in between, where they're potentially sort of just well enough to go out and about, and do things for themselves, but maybe things at home aren't good and they're not coping, and actually, as a GP, maybe it's really difficult to pick up on that. (General Practitioner_1)

### Impact on and from dementia (when managing physical health)
#### Inter-relatedness of cognition and physical health

As the outcomes associated with physical health and cognition were found to be inter-related, healthcare professionals sought to optimise physical and preventative healthcare in ways that improved cognition.

Yes, so, I think, you know, the physical health of someone has a really big impact on how they're dementia… they experience their dementia. So, they might just have a really poor diet which means that… or they might not be able to manage their diabetes and therefore they're feeling rubbish all the time. (Memory Service Manager)

Physical problems as well. You know? Because they have, not only possible dementia, but they also may have physical illnesses. Parkinson's, stroke, any sort of medical condition, as well, on top. So, we don't just concentrate on the dementia. We concentrate on the person. And it's a very much a holistic approach. (Occupational Therapist)

#### Limitations of physical health treatments in people with dementia

Some treatments prescribed for long-term conditions were considered to be unhelpful for people with dementia, for example because the strategies they used depended on memory and could not or had not been adapted to the context of dementia. In this next quote, a family carer explains how a person with dementia was too impaired to benefit from suggested strategies:

Well, she did have a visit from an occupational therapist [following a stroke]. They arranged that for a few weeks but he, it was when she was at her most florid time and he was recommending that she did things like writing a list down of things that interested her and what she would like to do and things like that. But she wouldn't do any of those. And he said, you know, he came about four or five times, no point in me coming because she doesn't take any notice of what I say, or do anything. (Family Carer_3, Sister)

Healthcare professionals and homecare workers sometimes felt that family carers were unable to accept that available treatments may not be helpful to people with dementia. They reported that this sometimes led to the continuation of treatments of questionable benefit, even when it went against the preferences of the person with dementia receiving them:

I suppose family members are sometimes trying to keep people as they would see them, well and

physically well for as long as possible and doing these things that their diabetes or anything else, and so they'll want them to do certain things whereas that's no longer what is important to that person. (General Practitioner_2)

I have a lady that's quite old and her mother, she wants the physio in and it's just not possible. She physically can't do it, but she thinks it'll keep her strong a little bit. And it's really not doing her any good, even the physio said that. But the daughter wants it. (Homecare Worker_2)

The above quotes highlight how the voice of people living with dementia can be lost when determining the benefits of treatment. Family carers appear to adopt an 'all or nothing' approach to physical health decision-making due to the lack of flexible interventions which successfully account for dementia.

At other times, family carers and professionals discussed and agreed approaches in collaboration with people with dementia. One family carer described how this was helpful where difficult, end-of-life decisions, balancing comfort with quality of life needed to be taken:

Kidney is only about 6% working. So because of that, you know… And we've decided not to go on dialysis. Not just we, but the specialist kidney doctor, we went to see him a couple of months ago and he suggested as well not to put him on dialysis because of his age… And he didn't think it was going to be beneficial to him… (Family Carer_5, Daughter)

## DISCUSSION
### Main findings
We describe three themes exploring how the self-management of comorbid long-term conditions is experienced and negotiated by people with dementia and their carers.

The first theme focused on the transition of roles which occurred in relation to advancement of dementia symptoms, with support increasing in accordance to level of need. Substituting self-management, especially medication, presented conflicts for family carers and homecare staff, to balance involving people with dementia in their care with safety concerns. The second theme identified communication as a key concept in the management of long-term conditions in dementia. Partnership working between people with dementia and family carers sought to overcome communication difficulties as a result of dementia. Without a family carer, changes in an individual's ability to self-manage may be difficult to detect, especially when people with dementia infrequently accessed primary care. The final theme highlighted the inter-relatedness of cognition and physical health and the importance of adapting physical healthcare to the context of dementia. Best interest decision-making is likely to be improved with increased awareness of likely harms and

benefits of treatment based on evidence such as age and stage of dementia.

Addressing the research gap recently identified in the field of dementia and multi-morbidity, these findings describe the interacting effects between cognition and physical health and provide an understanding of the care needs of this population.[4 21] In dementia care, family carers often experienced a role transition from a 'care-partner' (who provides assistance to a person who continues to manage much of their own care themselves) to 'caregiver' (who provides care) with increasing disability.[12] Our analysis demonstrates the critical role of family carers in communicating with primary care.[13] In addition to impacting memory, judgement and orientation, the impact of dementia on language skills has previously been identified as a barrier to reporting symptoms in healthcare appointments.[9] Relevant theoretical models of independence at home in dementia suggest that professionals should also be viewed as partners, rather than experts, when supporting self-management.[27] The integrated logic of care model suggests that psychosocial and physical needs have a great influence on each other, and should therefore be addressed simultaneously.[28] Our analysis highlights a similar need for physical health and cognition.

One of the main self-management tasks across long-term conditions is medication adherence.[8 9] Similar to the findings of Rapaport et al,[22] our analysis highlights the role of homecare workers in adapting a facilitative approach; specifically 'doing with not for' during medication management. The need for both family carers and homecare workers to prevent harm often conflicts with the desire of a person with dementia to remain autonomous.[29] The balancing of roles is complex for carers especially involving boundaries between advocating for independence and the implementation of medical advice.[30] Our findings demonstrate this dilemma extends to the management of long-term conditions in dementia. Future research is required to understand how calculated risks in care decision-making can enable freedom for people with dementia while managing carer safety concerns.[27] Living well with dementia has been conceptualised as living with quality of life, choice, autonomy, dignity and as independently as possible. There can be a tension between independence as an expression of autonomy and interdependence that can enable people with dementia to live in their own homes for longer. People with dementia and carers often have different goals and priorities for care. Judging at which point a person loses capacity to refuse care and when to intervene in their best interests can be challenging. Research has indicated the need for professionals to facilitate conversations around the decision to move to care homes.[31] A similar approach may be required during physical health decision-making.

One of the five core self-management skills is decision-making[8] impacted by dementia due to impairments in executive functioning.[9] Non-dementia health decisions, such as end-of-life care, have previously been identified

as a problematic decision-making area for family carers.[32] In this context, care decisions involving all stakeholders require a focus on quality of life over a target-driven approach[16] which is continuously reassessed according to changes in disease state.[17] Quality of life is an important indicator of effectiveness for person-centred care interventions in dementia.[33]

## Clinical implications

These findings highlight that management of long-term conditions in dementia exists on a continuum with support from all stakeholders developing across stages of dementia and severity of long-term conditions. To best support people with dementia to manage long-term conditions, treatments should be congruent to need and consider the impact of dementia on engagement. The development of dementia-specific consultation models must consider the impact of dementia on language skills and communication ability, and work with family carers acting in dyadic care relationships with people with dementia. Greater considerations of how healthcare professionals communicate the limitations of treatments that are not in the best interest or of questionable benefit to people with dementia and family carers would improve physical health decision-making. Such discussions are often complex, involving various stakeholders and ensuring the voice of people with dementia is heard when deciding on treatment benefits can be challenging. Service development must identify mechanisms for detecting change in self-management ability when family carer or homecare workers are not involved in supporting self-management at home for example, alerts for appointment non-attendance and medication underutilisation.

## Strengths and limitations

To our knowledge, this paper presents the first qualitative analysis to consider how people with dementia can be best supported to manage long-term conditions from the perspective of all stakeholders. Specifically, this paper expands on previous research by considering the role of the homecare worker in the care network, previously described in the literature as a triad between the person with dementia, their family carers and healthcare professionals.[34 35] The analysis includes interviews from a large and varied sample from a range of locations across the UK.

We used secondary analysis to explore sections of transcripts that discuss multi-morbidity in dementia. A limitation of this analytic approach is that the data were originally collected to explore how to support independence at home in dementia. Nevertheless, participants spontaneously discussed experiences of long-term conditions due to their salience, enabling further exploration of multi-morbidity in this analysis. All family carers explicitly discussed the management of long-term conditions which may account for large representation from this stakeholder in the analysis, while people with dementia, healthcare professionals and homecare workers would discuss more generally about medication management and physical health.

We included coauthors involved in original data collection (AB, ML, JB-D, RHG) to bring an inside knowledge of context during analysis to overcome the problem of the lead author 'not having been there'.[24] In addition, the lead author listened to recordings of transcripts to become familiar with data. The use of additional methods such as participant observations in future research would enable triangulation of results on how stakeholders can best support people with dementia to manage long-term conditions.[36]

We purposively recruited a diverse sample in terms of ethnicity and gender, and while we considered during analysis how these factors might intersect our themes, unlike in our previous publications from this database on different topics,[22 23] we did not find any clear areas where themes were gendered or were reflected differently across ethnic groups or immigration status.[37] While this was part of the reflective analytic process, we did not include an intersectionality analysis as an a priori analytic step. Williams *et al* did this, and we will consider adapting this approach in our future work.[38]

## CONCLUSION

Family carers and homecare workers support the management of long-term conditions in the home. Family carers act as proxies for communication in healthcare appointments. Healthcare professionals have a responsibility to identify changes in self-management ability as dementia progresses and adapt care accordingly. This process of substituting self-management evolves with advancement of dementia symptoms and relies on communication in the care network, while considering the impact on and from dementia to achieve holistic physical health management. Care decisions must consider people with dementia as a whole, and be based on realistic outcomes and best interests.

**Contributors** All authors made a substantial contribution to this work. JR, CC, KW and AB all contributed to the conception and design of the study and JR drafted the paper. All authors critically revised it and gave final approval for this version to be published. AB, ML, JB-D, PR and RHG collected the qualitative data. JR, CC, KW and AB all coded some of the interview transcripts. JR then organised the data into preliminary themes and led the subsequent analysis. CC, KW and AB further refined the thematic analysis in an iterative process.

**Funding** This work was supported by the Economic and Social Research Council [grant number ES/P0 0 0592/1]; and the National Institute of Health Research (NIHR) Collaboration for Leadership and Applied Health Research (North Thames). Data collection was carried out within the UCL Alzheimer's Society Centre of Excellence for Independence at home, NIDUS (New Interventions in Dementia Study) programme (Alzheimer's Society Centre of Excellence grant 330).

**Disclaimer** The views expressed are those of the author(s) and not necessarily those of the NIHR or the Department of Health and Social Care.

**Competing interests** None declared.

**Patient and public involvement** Patients and/or the public were involved in the design, or conduct, or reporting, or dissemination plans of this research. Refer to the Methods section for further details.

**Patient consent for publication** Not required.

**Ethics approval** London (Camden and Kings Cross) Research Ethics Services approved the study in November 2017 (reference: 17/LO/1713).

**Provenance and peer review** Not commissioned; externally peer reviewed.

**Data availability statement** All data relevant to the study are included in the article or uploaded as supplemental information. The qualitative data used and analysed during the current study are available from the corresponding author on reasonable request.

**ORCID iDs**
Jessica Laura Rees http://orcid.org/0000-0002-9471-2134
Kate R Walters http://orcid.org/0000-0003-2173-2430

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
