## [Reviewer comments · BMJ Open]

ARTICLE DETAILS

TITLE (PROVISIONAL)	Exploring how people with dementia can be best supported to manage long-term conditions: A qualitative study of stakeholder perspectives
AUTHORS	Rees, Jessica Laura; Burton, Alexandra; Walters, Kate R; Leverton, Monica; Rapaport, Penny; Herat Gunaratne, Ruminda; Beresford-Dent, Julie; Cooper, Claudia

VERSION 1 – REVIEW

REVIEWER	Martina Roes German Center for Neurodegenerative Diseases Germany
REVIEW RETURNED	03-Jul-2020

GENERAL COMMENTS	Why was the systematic review of theoretical models and RCT evidence (Lord et al. 2019, online) not included? if a study protocol for the NIDUS study exist, the reference should be included Since the sample has an ethnicity component (13 family caregivers, 6 staff) this theme needs to be addressed (introduction, recruitment, interpretation of the results, discussion) e.g. Rote & Moon (2016) racial/ethnic differences in caregiving frequency // e.g. Williams et al. (2016) A Canadian qualitative study exploring the diversity of the experience of family caregivers ... It may need to be clarified if the included people are immigrants or born in the UK. the relation of the family caregiver: pls add in table clarify the term 'care-partner' / 'care-giver' It's of interest for the reviewer / reader to add the Nivo code tree (as a supplement, which then would be equal to data analysis in quantitative studies provided by statisticians)
---

REVIEWER	Christine Stirling University of Tasmania Australia
REVIEW RETURNED	16-Jul-2020

GENERAL COMMENTS	This is an important topic and I believe using secondary data is an important value add. The paper is well written in clear language though the quotes could be shortened in the findings section as they were at times unnecessarily long. I felt though that the discussion
---

	and conclusion needed some additional work. Discussion: the first paragraph basically lists all the findings in a long and repetitive first paragraph. I feel this should be deleted and replaced with a more conceptual overview paragraph around the three key themes. In the remainder of the discussion I believe more attention should be drawn to the tension between autonomy and human rights and decreasing cognitive capacity - how is capacity assessed, at which point is refusal of care a symptom of cognitive decline. The tension between carer values and the values of the person with dementia and quality of life. Conclusion: in the abstract conclusion and the conclusion I feel the need to move from self-management to substituting self-management as cognitive capacity decreases needs to be made explicit. A further small point - on page 9 when discussing NOMAD in quotes an explanation in brackets would help international audiences. Overall an interesting contribution to the literature.
--	---

VERSION 1 – AUTHOR RESPONSE

Reviewer 1

Why was the systematic review of theoretical models and RCT evidence (Lord et al. 2019, online) not included?

The reference for the NIDUS theoretical model has been added to the discussion (Page 12, Line 33)

If a study protocol for the NIDUS study exist, the reference should be included

No published protocol for the qualitative component of NIDUS exists. However, publically available study details (ISRCTN registration numbers) for the NIDUS feasibility and main trials have been added to the methods section (Page 4, Line 2 and 3).

Since the sample has an ethnicity component (13 family caregivers, 6 staff) this theme needs to be addressed (introduction, recruitment, interpretation of the results, discussion) e.g. Rote & Moon (2016) racial/ethnic differences in caregiving frequency // e.g. Williams et al. (2016) A Canadian qualitative study exploring the diversity of the experience of family caregivers ...

Thank you for pointing us towards this interesting paper. Williams et al. (2016) employed a multi-step qualitative analysis, in which an intersectionality analysis was included as a second and third step to finalise themes, to explore how participants' social locations such as age, education, gender, geography, ethnicity, etc. influenced findings. We did not do this as an a priori analytic step, so have acknowledge this in the limitations (Page 15, Line 4-10).

The experiences of South Asian carers analysed using the same dataset has previously been published by the NIDUS team:

<https://academic.oup.com/gerontologist/article/60/2/331/5581953>

We have added a more explicit reference to this paper in the methods section (Page 4.Line 6).

It may need to be clarified if the included people are immigrants or born in the UK.

This information is available in a previously published paper from the same dataset: Herat-Gunaratne et al (2019)<https://academic.oup.com/gerontologist/article/60/2/331/5581953> and we have also added it to the results section (Page 5, Line 23).

Based on the findings of Rote & Moon (2016), we did not consider immigration status during our analysis, and have added this to the limitations section (Page 15, Line 7).

the relation of the family caregiver: pls add in table

The relationship of the family caregiver to the person living with dementia has now been added to Table 1 (Page 5, Line 25).

clarify the term 'care-partner' / 'care-giver'

The terminology 'care-partner' compared to 'care-giver' has been clarified in the Discussion section (Page 12, Line 28-29)

It's of interest for the reviewer / reader to add the Nivo code tree (as a supplement, which then would be equal to data analysis in quantitative studies provided by statisticians)

The NvivoCoding Framework has been added as a supplement to this article.

Reviewer 2

This is an important topic and I believe using secondary data is an important value add. The paper is well written in clear language though the quotes could be shortened in the findings section as they were at times unnecessarily long. I felt though that the discussion and conclusion needed some additional work.

Thank you for your feedback. We have shortened the quotes in the findings section and address your comments on the discussion and conclusion below.

Discussion: the first paragraph basically lists all the findings in a long and repetitive first paragraph. I feel this should be deleted and replaced with a more conceptual overview paragraph around the three key themes.

The first paragraph of the discussion (Page 12) has been amended to present a conceptual overview of key themes. The text now reads:

"The first theme focused on the transition of roles which occurred in relation to advancement of dementia symptoms, with support increasing in accordance to level of need. Substituting self-management, especially medication, presented conflicts for family carers and homecare staff, to balance involving people with dementia in their care with safety concerns. The second theme identified communication as a key concept in the management of long-term conditions in dementia. Partnership working between people with dementia and family carers sought to overcome communication difficulties as a result of dementia. Without a family carer, changes in an individual's ability to self-manage may be difficult to detect, especially when people with dementia infrequently accessed primary care. The final theme highlighted the inter-relatedness of cognition and physical health and the importance of adapting physical healthcare to the context of dementia. Best interest decision making is likely to be improved with increased awareness of likely harms and benefits of treatment based on evidence such as age and stage of dementia."

In the remainder of the discussion I believe more attention should be drawn to the tension between autonomy and human rights and decreasing cognitive capacity - how is capacity assessed, at which point is refusal of care a symptom of cognitive decline. The tension between carer values and the values of the person with dementia and quality of life.

We have added to the discussion (Page 13), the text now reads:

"Living well with dementia has been conceptualised as living with quality of life, choice, autonomy, dignity and as independently as possible. There can be a tension between independence as an expression of autonomy and interdependence that can enable people with dementia to live in their own homes for longer (Lord et al., 2019). People with dementia and carers often have different goals and priorities for care. Judging at which point a person loses capacity to refuse care, and when to intervene in their best interests can be challenging."

Conclusion: in the abstract conclusion and the conclusion I feel the need to move from self-management to substituting self-management as cognitive capacity decreases needs to be made explicit.

We have amended each conclusion section (Page 15, Line 16) to explicitly discuss the implications of changes in cognitive capacity to self-management ability.

A further small point - on page 9 when discussing NOMAD in quotes an explanation in brackets would help international audiences.

The term 'nomad' has been defined as suggested for international audiences. The text now reads: "... it has to be in a nomad [pre-packed medication]"

VERSION 2 – REVIEW

REVIEWER	Martina Roes DZNE/Witten site Germany
REVIEW RETURNED	03-Aug-2020

GENERAL COMMENTS	The improvement of the updated version is quite satisfactory!!!
---

REVIEWER	Christine Stirling University of Tasmania
REVIEW RETURNED	21-Aug-2020

GENERAL COMMENTS	An interesting contribution of the literature
---